# Safety of Chuna Manipulation Therapy in 289,953 Patients with Musculoskeletal Disorders: A Retrospective Study

**DOI:** 10.3390/healthcare10020294

**Published:** 2022-02-02

**Authors:** Suna Kim, Gook-Beom Kim, Hyo-jun Kim, Joon Park, Ji-Won Lee, Wu-jin Jeong, Hye-Gyeong Kim, Min-Young Kim, Kyoung-Sun Park, Jinho Lee, Jun-Hwan Lee, Joon-Shik Shin, Byung-Cheul Shin, In-Hyuk Ha

**Affiliations:** 1Daejeon Jaseng Hospital of Korean Medicine, Daejeon 35263, Korea; tnsdk2648@jaseng.org (S.K.); gbkim89@naver.com (G.-B.K.); nobel1990@daum.net (H.-j.K.); haspark@daum.net (J.P.); ljwsky535@naver.com (J.-W.L.); mysisi@naver.com (W.-j.J.); khk3629@naver.com (H.-G.K.); 2Jaseng Spine and Joint Research Institute, Jaseng Medical Foundations, Seoul 06110, Korea; ann2310@naver.com; 3Jaseng Hospital of Korean Medicine, Seoul 06110, Korea; lovepks0116@gmail.com (K.-S.P.); jhl3006@gmail.com (J.L.); jasengmaster@gmail.com (J.-S.S.); 4Clinical Research Division, Korea Institute of Oriental Medicine, Daejeon 34054, Korea; omdjun@kiom.re.kr; 5Korean Medicine Life Science, Campus of Korea Institute of Oriental Medicine, University of Science & Technology (UST), Daejeon 34113, Korea; 6School of Korean Medicine, Pusan National University, Yangsan 50612, Korea; 7Spine & Joint Center, Pusan National University Korean Medicine Hospital, Yangsan 50612, Korea

**Keywords:** Chuna manipulation therapy, manipulation, manual therapy, safety, adverse events, musculoskeletal disorders, Korean medicine

## Abstract

Studies have reported that mild adverse events (AEs) are common after manual therapy and that there is a risk of serious injury. We aimed to assess the safety of Chuna manipulation therapy (CMT), a traditional manual Korean therapy, by analysing AEs in patients who underwent this treatment. Patients who received at least one session of CMT between December 2009 and March 2019 at 14 Korean medicine hospitals were included. Electronic patient charts and internal audit data obtained from situation report logs were retrospectively analysed. All data were reviewed by two researchers. The inter-rater agreement was assessed using the Cohen’s kappa coefficient, and reliability analysis among hospitals was assessed using Cronbach’s Alpha coefficient. In total, 2,682,258 CMT procedures were performed in 289,953 patients during the study period. There were 50 AEs, including worsened pain (*n* = 29), rib fracture (*n* = 11), falls during treatment (*n* = 6), chest pain (*n* = 2), dizziness (*n* = 1), and unpleasant feeling (*n* = 1). The incidence of mild to moderate AEs was 1.83 (95% confidence interval [CI] 1.36–2.39) per 100,000 treatment sessions, and that of severe AEs was 0.04 (95% CI 0.00–0.16) per 100,000 treatment sessions. Thus, AEs of any level of severity were very rare after CMT. Moreover, there were no instances of carotid artery dissection or spinal cord injury, which are the most severe AEs associated with manual therapy in other countries.

## 1. Introduction

Chuna manipulation treatment (CMT) is a form of manual therapy used by Korean medicine doctors (KMDs) to treat structural or functional pathologies by applying effective stimuli to a patient’s body using hands or an assistive device. It was developed by employing and advancing various manual therapy strategies from other countries on the basis of traditional Chuna therapy, which is rooted in Korean medicine theories that emphasise a balance between function and structure. The techniques employed in CMT are divided into two broad types: bonesetting Chuna therapy that is performed on the joints in the body, and fascia Chuna therapy that is performed on the soft tissues. Bonesetting Chuna therapy was advanced from chiropractic therapy and encompasses techniques such as manipulation; high-velocity, low-amplitude technique; and thrust. Fascia Chuna therapy was advanced from Chinese tui na and encompasses joint mobilisation, joint distraction, and soft tissue technique [1]. CMT has been incorporated into the Korean health care system and has been administered in 16.4% of inpatients and 83.6% of outpatients with musculoskeletal disorders in Korean medicine hospitals specialising in spine and joint diseases [2]. Since 2019, health insurance coverage in Korea has been expanded to include CMT for the treatment of musculoskeletal disorders [2].

Manual therapy is performed in various forms by chiropractors, osteopaths, and physical therapists across the world, including the United States, Europe, and Australia. The use of spinal manipulation has increased in recent decades in Western countries [3], as has the popularity of chiropractic therapy among American adults [4]. The UK National Institute for Health and Clinical Excellence guidelines now recommend manual therapy for treating persistent or subacute lower back pain [5,6]. Several studies have found that osteopathic care and spinal manipulation therapy are effective for reducing pain and promoting functional recovery in patients with musculoskeletal disorders, particularly those involving the spine [7,8,9]. However, the safety of CMT remains controversial. Previous reports have suggested that CMT-related adverse events (AEs) are rarely severe [10,11]; nonetheless, some have argued that manual therapy, especially cervical spine manipulation, poses an unacceptable risk because of the possibility of severe AEs, such as carotid artery dissection and spinal cord injury [12].

The reported incidence of AEs associated with manual therapy varies from 0.00005% to 83% depending on the research methodology used and the severity of the AEs [10]. To date, there has been no large-scale study on the safety of manual therapy. Considering the growing worldwide popularity of manual therapy [3], more data on safety are needed, including for CMT. The aim of this multicentre study was to assess the safety of CMT in patients receiving this therapy at Korean medicine hospitals.

## 2. Materials and Methods

This study is a retrospective analysis of electronic patient records and internal hospital audit data drawn from event report logs at 14 Korean medicine hospitals located in major cities of the Republic of Korea.

### 2.1. Study Population and Data Collection

Our institution has been designated as a “spine specialty Korean medicine hospital” by the Korean Ministry of Health and Welfare and has 20 branches in main centres nationwide. For this study, we excluded centres with an incomplete AE reporting system, and collected data from 14 Jaseng hospitals located in Gangnam, Bucheon, Jamsil, Gwanghwamun, Nowon, Ansan, Suwon, Incheon, Daejeon, Cheongju, Gwangju, Daegu, Haewundae, and Ulsan. The target study population comprised inpatients and outpatients with musculoskeletal disorders who received at least one session of CMT between December 2009 and March 2019. Inpatient treatment was performed if the patient was in severe pain that interfered with daily life or if outpatient treatment was not possible. The data collected from the electronic records included patient age and sex, district, disease or injury, number of CMT sessions received, length of hospital stay, and number of hospitalisations. AEs were identified in situational report logs; when an AE was confirmed to be associated with CMT, additional data were obtained from relevant records (e.g., progress notes, imaging or blood test results, the nurse’s situation report log, and patient complaints). The patients’ ID was anonymised for privacy of the data. All documents were encrypted, and only authorised researchers had accessed to the documents.

### 2.2. Nursing Situation Reports and Routine Vigilance by Medical and Nursing Staff

Nurses are trained to record and report all AEs, complaints, and other noteworthy issues on a daily basis. These records are collated weekly and submitted to the head nurse and KMDs. Nurses perform at least five daily rounds for inpatients and attend outpatients at least twice per visit. Therefore, any untoward event is easily detected and well documented. Events occurring outside the hospital can be reported by the patient or caregiver to the on-call KMD via a hospital landline that is functional for 24 h a day and are recorded in the nursing department. Furthermore, nurses routinely contact patients who stop treatment prematurely by telephone as part of the clinical follow-up to ensure the quality of care. Further, the patients are instructed to notify the hospital via the 24-h nurse line in case of relapse of pain or in the event of any other AEs even after the discontinuation of treatment.

### 2.3. Screening for AEs

Nursing situation reports from the 14 Korean medicine hospitals were integrated into our data. Reports containing inadequate data were excluded. Ten KMDs with at least 3 years of clinical experience received offline education about the definition, criteria, and screening for AEs. Additional educational materials were provided as needed during the screening phase.

The screening was performed in two stages. First, the 10 KMD researchers were allocated to screen the nursing situation reports and identify AEs associated with CMT in such a way that all data were double-checked by two researchers. If one researcher in a pair determined an AE to be associated with CMT, the screening proceeded to the second stage, in which the two researchers reviewed pertinent comorbidities, results of AE-related investigations, and medical records to determine causality and the severity of the AE, referencing the criteria proposed by the World Health Organization-Uppsala Monitoring Centre (WHO-UMC) causality scale and National Cancer Institute [13,14]. Any disagreement was resolved by an additional pair of specialists (Figure 1).

### 2.4. Analysis of AEs

An AE was defined as any unfavourable sign, symptom, or disorder temporally associated with the treatment, irrespective of whether the relationship was deemed causal [15], in accordance with the International Conference on Harmonisation [16].

The severity of each AE was classified according to the National Cancer Institute guideline as mild, moderate, severe, life-threatening, or fatal [13] (Grade 1: mild; asymptomatic or mild symptoms; clinical or diagnostic observations only; intervention not indicated. Grade 2: moderate; minimal, local, or non-invasive intervention indicated; limiting age-appropriate instrumental activities of daily living (ADL). Grade 3: severe or medically significant but not immediately life-threatening; hospitalisation or prolongation of hospitalisation indicated; disabling; limiting self-care ADL. Grade 4: life-threatening consequences; urgent intervention indicated. Grade 5: death related to AE.)

Causality was determined using the WHO-UMC causality scale, and the terminology was revised as appropriate for CMT. Causality was assessed as certain, probable, possible, or unlikely (Appendix A) [14].

### 2.5. Patient and Public Involvement

Patients were involved in reporting AEs of CMT and their follow-up result.

### 2.6. Statistical Analysis

Inter-rater agreement at each hospital was assessed using the Cohen’s kappa coefficient [17]. Reliability analysis among hospitals was assessed using Cronbach’s Alpha coefficient. The incidence of AEs was recorded as events per 100,000 person-days and per 100,000 treatments, and the incidence was classified according to the European Commission guidelines for AEs involving medicinal products [18]. The confidence interval of the incidence rate was calculated with Poisson distribution. R version 4.1.1 (© The R Foundation for Statistical Computing) was used in our analysis.

## 3. Results

### 3.1. Baseline Characteristics

Between December 2009 and March 2019, 47,965 inpatients received 807,103 CMT sessions and 241,988 outpatients received 1,875,155 CMT sessions at the 14 participating institutions, resulting in 2,682,258 CMT sessions for 289,953 patients (Table 1). The most common indication for CMT was S134 (sprain or strain of the cervical spine, 19%), followed by S335 (sprain or strain of the lumbar spine, 18%) and M511 (lumbar or other intervertebral disc disorder with radiculopathy, 11.9%; Table 2).

### 3.2. Inter-Rater Agreement and Reliability Analysis

A total of 3759 patient AE reports were independently analysed by two researchers to determine the frequency of CMT-associated AEs. The inter-rater agreement at the hospitals located in Daegu, Jamsil, Cheongju, Suwon, and Incheon was perfect, and it was 0.877 at the other hospitals. The mean Cohen’s kappa coefficient for all hospitals was 0.9805 (Appendix A). Cronbach alpha coefficient among hospitals was 0.58 (95% 0.32–0.83).

### 3.3. Types and Frequency of AEs

CMT-associated AEs included worsened pain (*n* = 29; Appendix A), rib fracture (*n* = 11; Appendix A), fall during treatment (*n* = 6; Appendix A), chest pain (*n* = 2; Appendix A), dizziness (*n* = 1), and an unpleasant sensation (*n* = 1; Appendix A). A total of 50 AEs were associated with 2,682,258 CMT sessions, of which only one was deemed to be serious (avascular necrosis of the femoral head followed by surgery). The overall incidence of AE was 1.86 per 100,000 treatment sessions (Table 3).

#### 3.3.1. Worsening of Musculoskeletal Pain

Most cases of worsening pain involved radiation of pain in the lumbar region and lower limbs (*n* = 23), with additional reports of cervical pain (*n* = 2), knee-related symptoms (*n* = 2), worsening hip pain (*n* = 1), and the onset of temporomandibular joint pain (*n* = 1). There were 15 cases under the age group of 40 years and younger, 10 in the age group of 4059, and 4 in the age group of 60 years and older. The mean number of CMT sessions was 9.3. Twenty-one of the 23 patients with worsening of radiating lumbar or lower limb pain had pre-CMT magnetic resonance imaging (MRI) scan results available, the findings of which were abnormal in all cases. Twelve of these patients showed disc extrusion and seven showed disc protrusion. Pre-procedural MRI showed disc herniation in the two patients with worsened neck pain. Nine patients in whom the worsened lumbar pain did not improve were referred for a nerve block with or without continuation of their existing treatment. Eight patients continued with their Korean medicine regimen even after the worsening of their lumbar pain. Nine patients had a mild increase in musculoskeletal pain, which did not require additional treatment. Further measures, such as nerve block or analgesics, were required in 19 moderate cases, and one patient with severe worsening pain was referred to the surgeons upon diagnosis of avascular necrosis (AVN) of the femoral head. Causality was considered possible in 28 cases and probable in one.

#### 3.3.2. Rib Fractures

All 11 rib fractures following CMT were confirmed on radiographs, bone scans, or ultrasound scans. All fractures occurred in female patients. The minimum age of onset was 54 years; nine cases were over 60 years of age, with a median age of 70 years. All fractures were rated as moderate. Causality was deemed to be certain in nine cases (82%). Six of the 11 women had also been using medicinal herbs at the time of the fracture.

#### 3.3.3. Falls during Treatment

The minimum age of patients who experienced a fall was 54 years, and the median age was 63.5 years. The falls occurred during climbing onto the Chuna bed, changing position on the bed, or leaving the bed. Four patients sustained a contusion as a result of their fall but had no symptoms. Pre-emptive radiographs were obtained in 50% of patients, none of whom required additional treatment; the contusions were rated as mild in these cases.

#### 3.3.4. Vertigo, Chest Pain, and Unpleasant Sensation

Other AEs included one case each of an unpleasant sensation and headache with vertigo and two of chest pain. The patient who reported an unpleasant sensation was a 26-year-old woman who received CMT in the supine position while wearing a skirt. The 38-year-old patient who complained of headache and vertigo immediately after CMT had no abnormal brain MRI or MR angiography findings, and the symptoms improved after using herbal medicine and anti-inflammatory drugs for five days. Two patients who had chest pain were in their 70s and had no abnormal radiographic or ultrasound findings, and the pain had improved 1 month later.

## 4. Discussion

Although manual therapy is effective for musculoskeletal disorders [19], there have been reports of potentially severe AEs, including carotid artery dissection, cerebral infarction, dural tear, and cauda equina syndrome [1,20]. However, the previous populations studied were not sufficiently large to establish causality with a high degree of reliability [10,21]. Moreover, the experts performing the manual therapy were too diverse to be able to collaborate on systematic data collection and reporting of AEs for a large patient population [22]. In this study, we examined all CMT-related AEs in a very large patient population in Korea to determine the incidence of AEs and the safety of CMT.

Although exacerbation and progression of disc herniation following manual therapy have been reported [10], a study by Oliphant found this to be rare, at fewer than 1 case per 3.7 million treatments [23]. Given that all patients who present to a Jaseng hospital with symptoms of cauda equina syndrome or exacerbation of disc herniation undergo prompt MRI, it is unlikely that radiological findings indicating serious exacerbation were overlooked. However, considering that the symptoms of disc herniation tend to wax and wane [24] and that acupuncture, pharmacopuncture, and herbal therapies are often performed concurrently with CMT, the exact causality could not be established in our patients with worsening pain. Nevertheless, the increase in pain was transient in most patients (Appendix A).

Kranenburg et al. [25] analysed 144 studies of AEs associated with manual therapy involving the neck and found cervical arterial dissection to be the most common AE (57%; *n* = 129/227). They also found that 66% of patients who developed an AE had been treated by a chiropractor. In the present study, there were two cases of worsened neck pain; however, no life-threatening or severe AEs were observed that would support previous concerns [12,20]. Furthermore, none of the patients with increased pain after CMT had a dural tear or spinal cord injury, which has been reported previously [23]; this may reflect the fact that KMDs routinely perform imaging to check for underlying conditions that may be a contraindication to manual therapy or increase the risk for AEs. The only severe AE in our study occurred in a patient who developed hip pain after CMT and was found to have AVN of the femoral head that required surgery. Since the pre-CMT radiograph indicated AVN, it is difficult to conclude that CMT was the cause; however, as the patient experienced increased pain, the possibility that CMT aggravated the AVN cannot be excluded.

There were clear sex- and age-specific characteristics in the 11 cases of rib fractures. Female sex hormones play a crucial role in maintaining bone mineral density [26]; therefore, although menopause is not necessarily a contraindication, postmenopausal women require careful monitoring during joint manipulation [1]. Women with a low bone mineral density have an elevated risk for rib fractures [27]. It is likely that most of these fractures occurred during thoracic correction techniques. Changes in the thoracic curvature tend to induce secondary changes in the cervical and lumbar curvatures [1]. Therefore, it is likely that CMT was performed in the thoracic area in patients complaining of problems in the lumbar and cervical areas. When correcting the lower thoracic spine with the patient in the prone position, the practitioner places the hands on the transverse processes and applies pressure in the posterior to anterior direction; rib trauma may occur if the force applied is not in the correct direction.

All six cases of falls resulted in mild contusions that did not require additional treatment. The median age of the patients who experienced falls was 63.5 years, which is expected given that falls are more common in older patients. Approximately 35–40% of individuals aged ≥65 years fall at least once a year [28]. All our CMT-related falls occurred while using a table specifically designed for CMT and could be prevented by closer supervision of patients at high risk for falls.

One patient had headache and vertigo that persisted for 5 days after CMT, but mild and transient discomfort following CMT, such as systemic pain and headache, may be interpreted as a normal response to mobilisation or stimulation of periarticular soft tissue [1]. One patient in our study complained of an unpleasant sensation during CMT, which was because of receiving CMT while wearing a skirt. If the patient’s apparel is not suitable for undergoing treatment, either a gown should be available, or the patient should be informed and sign an informed consent form in advance to prevent such AEs.

We identified 49 cases of mild to moderate AEs, with an incidence rate of 1.83 per 100,000 treatments, which was lower than that (2.5) reported by a study that retrospectively examined chiropractic safety [11]. There was only one case of severe AEs, with an incidence rate of 0.04 per 100,000 treatments, which was also lower than that (0.05–0.25) reported previously [29]. This may be attributable to the fact that all CMT procedures were performed by the same group of specialists. Most KMDs who performed CMT in this study had completed 6 years of Korean medicine coursework, had passed the Korean national licensing examination for traditional Korean medicine, and had completed 4 years of hospital training; they were, therefore, trained in standardised Chuna technique and quality. These specialists had also completed regular coursework at the Korean Society of Chuna Manual Medicine for Spine and Nerves, which is a member of the International Federation for Manual/Musculoskeletal Medicine, as well as periodically completed mandatory refresher courses offered by the Association of Korean Medicine, which suggests that they were adequately familiar with the essential information on pre-CMT care. Second, CMT is an elaborate technique that localises and minimises the shock applied to the body. CMT practitioners are trained to perform joint mobilisation, joint distraction, and soft tissue techniques before performing manipulation (thrust), a high-velocity low-amplitude technique [1]. Hurwitz et al. [30] reported that the incidence of AE was higher with manipulation than with joint mobilisation, and it is possible that joint or soft tissue mobilisation and relaxation prior to manipulation lower the risk of AEs by preventing sudden delivery of shocks to the body.

This study is of value in that it included a markedly greater number of cases than those examined in past studies investigating AEs. Nielsen et al. targeted the largest study population of 34,605 in their overview of 125 studies to examine the safety of the spinal manual therapy [31]. In this study, 289,953 patients and more than 2.5 million cases of CMT were reviewed, making it a rare, very wide-ranging, and reliable investigation of severe AEs. Furthermore, by investigating all patients who underwent CMT in hospitals in various districts of Korea without applying strict inclusion and exclusion criteria, our findings reflect the real-life clinical situation. The hospitals in this study consistently controlled quality of care and were equipped with a systematic AE reporting system and trained personnel, which enabled large-scale AE monitoring. KMDs checked for AEs at every patient visit, and nurses communicated with patients before and after each treatment to follow up on their post-treatment status. As patients who stopped visiting the hospital were routinely contacted by KMDs and nurses by telephone, no patients were lost to clinical follow-up for AEs. Weekly event reports submitted by nurses contained objective records of even minor events, and based on these reports, the administration and medical teams periodically discussed each event and corrected relevant problems. 

One limitation of this study is its retrospective design, which means that mild AEs may have been missed and the incidence rate underestimated. However, AEs that require additional treatment or were abnormal would have been detected by KMDs; hence, it is highly unlikely that significant AEs were overlooked. Moreover, the retrospective design made it difficult to examine the incidence of AE according to each type of manual therapy technique. Another limitation is the association between AEs and CMT, given that treatments other than CMT, which had been administered in most patients, may have caused increased musculoskeletal pain. Acupuncture (100%), pharmacotherapy and herbal therapy (93%), and cupping therapy (83%) were performed concurrently with CMT in the patients who complained of increased pain; hence, it was impossible to attribute causality to CMT alone. We adopted a conservative approach in this study by including AEs with unclear causality. In addition, although standardised CMT was conducted through education and qualification tests, there may be differences in the level of each operator due to the nature of manual therapy.

This study found that severe AEs after CMT were very rare. In particular, there were no reports of carotid artery dissection or spinal cord injury, which are reportedly the most dangerous AEs associated with manual therapy as found by studies conducted in other countries. In our study, although inclusion/exclusion criteria or outcome did not include the pre-treatment imaging results, they were used for diagnosis and treatment planning by clinicians. Moreover, the researchers used the imaging results to determine a causal relationship in the event of an adverse reaction to CMT. Although recent evidence suggests that imaging should not be routinely used for most musculoskeletal disorders [32], to ensure patient safety, we performed routine imaging tests to check for fractures; malformations, such as os odontoideum; tumour; and joint dislocation and surgical status prior to applying techniques that target bones, such as the bonesetting Chuna therapy. As most mild AEs were preventable, the incidence of AEs could be further reduced by ongoing quality improvement in hospitals [30].

## 5. Conclusions

Our analysis of 289,953 patients and 2,682,258 cases of CMT indicates that both mild–moderate and severe AEs are rare after CMT. A total of 29 cases of increased musculoskeletal pain, 11 cases of rib fracture, 6 cases of falls, 2 cases of chest pain, 1 case of vertigo, and 1 case of mild discomfort were identified. The overall AE incidence was 1.86 per 100,000 treatment sessions. Serious AEs were identified in 1 out of 50 cases, with a frequency of 0.04 per 100,000 procedures. CMT performed by an expert can be considered relatively safe. Well-designed prospective studies are needed to examine CMT-related AEs in more detail.

## Figures and Tables

**Figure 1 healthcare-10-00294-f001:**
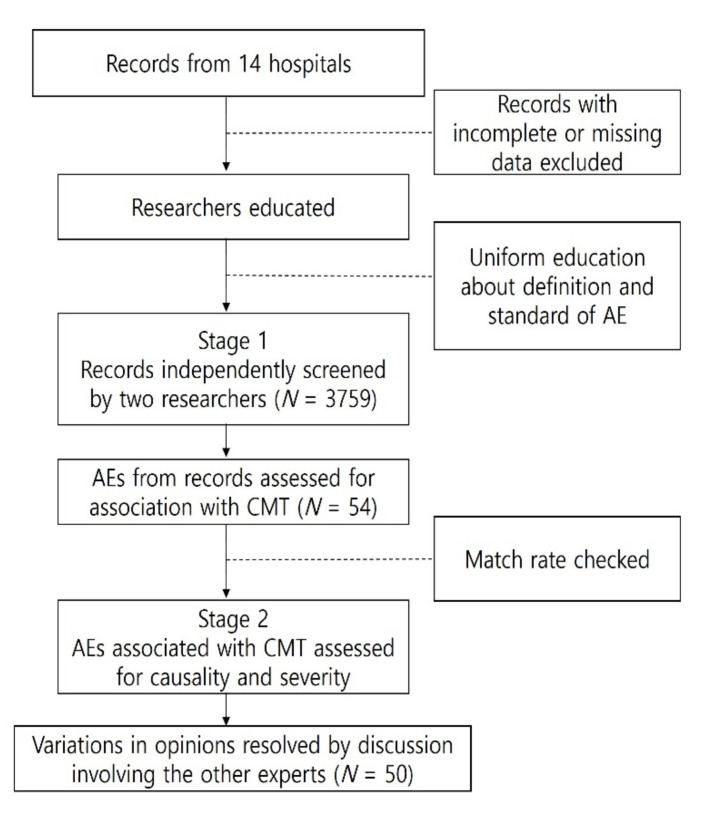
Flow chart showing the study protocol. AE, adverse event; CMT, Chuna manipulation therapy.

**Table 1 healthcare-10-00294-t001:** Demographic and clinical characteristics of patients who received Chuna manipulation therapy.

Characteristics	Total	Inpatient *	Outpatient
Patients, *n*	289,953	47,965	241,988
CMT sessions, *n*	2,682,258	807,103	1,875,155
Demographics			
Sex			
Female	145,955 (50.3)	24,531 (51.1)	121,424 (50.2)
Male	143,998 (49.7)	23,434 (48.9)	120,564 (49.8)
Age, years			
Mean ± SD	40.6 ± 13.8	42.3 ± 14.1	40.22 ± 13.74
0–9	825 (0.3)	70 (0.1)	755 (0.3)
10–19	8829 (3.0)	885 (1.8)	7944 (3.3)
20–29	55,624 (19.2)	8968 (18.7)	46,656 (19.3)
30–39	91,350 (31.5)	13,275 (27.7)	78,075 (32.3)
40–49	56,908 (19.6)	9389 (19.6)	47,519 (19.6)
50–59	45,164 (15.6)	9047 (18.9)	36,117 (14.9)
60–69	22,790 (7.9)	4760 (9.9)	18,030 (7.5)
70–79	7455 (2.6)	1342 (2.8)	6113 (2.5)
≥80	1008 (0.3)	229 (0.5)	779 (0.3)
District			
Seoul	157,686 (54.4)	14,271 (29.8)	143,415 (59.3)
Gyeonggi-do Province	68,696 (23.7)	10,658 (22.2)	58,038 (24.0)
Chungcheong-do Province	41,722 (14.4)	13,759 (28.7)	27,963 (11.6)
Gyeongsang-do Province	11,585 (4.0)	6486 (13.5)	5099 (2.1)
Jeolla-do Province	10,264 (3.5)	2791 (5.8)	7473 (3.1)
Treatment			
Admissions, *n*	-	1.1 ± 0.5	-
Duration of admission, days	-	15.0 ± 19.5	-
Outpatient visits	7.4 ± 11.4	5.3 ± 11.7 ^†^	7.8 ± 11.3
CMT sessions per patient, *n*	9.3 ± 13.5	16.8 ± 19.6	7.8 ± 11.3

Values are defined as numbers (percentages) or mean ± SD. * Defined as inpatient if hospitalised more than once during the observation period. ^†^ Additional outpatient visits after discharge. CMT, Chuna manipulation therapy; SD, standard deviation.

**Table 2 healthcare-10-00294-t002:** Most common indications for Chuna manipulation therapy *.

Total (*n* = 2,184,722)		*n* (%)
S134	Sprain and strain of cervical spine	415,270 (19.0)
S335	Sprain and strain of lumbar spine	393,292 (18.0)
M511	Lumbar and other intervertebral disc disorder with radiculopathy	260,068 (11.9)
M519	Intervertebral disc disorder, unspecified	137,949 (6.3)
M518	Other specified intervertebral disc disorder	112,307 (5.1)
M545	Low back pain	108,010 (4.9)
M501	Cervical disc disorder with radiculopathy	80,600 (3.7)
M542	Cervicalgia	72,236 (3.3)
M480	Spinal stenosis	68,988 (3.2)
M509	Cervical disc disorder, unspecified	53,349 (2.4)
Inpatient (*n* = 53,862) ^†^		
S134	Sprain and strain of cervical spine	14,468 (26.9)
S335	Sprain and strain of lumbar spine	12,446 (23.1)
M511	Lumbar and other intervertebral disc disorder with radiculopathy	7834 (14.5)
M501	Cervical disc disorder with radiculopathy	2870 (5.3)
M518	Other specified intervertebral disc disorder	2281 (4.2)
M480	Spinal stenosis	1663 (3.1)
S836	Sprain and strain of other and unspecified part of knee	1293 (2.4)
M545	Low back pain	1169 (2.2)
M519	Intervertebral disc disorder, unspecified	746 (1.4)
M255	Joint pain	589 (1.1)
Outpatient (*n* = 2,130,860) ^‡^		
S134	Sprain and strain of cervical spine	400,802 (18.8)
S335	Sprain and strain of lumbar spine	380,846 (17.9)
M511	Lumbar and other intervertebral disc disorder with radiculopathy	252,234 (11.8)
M519	Intervertebral disc disorder, unspecified	137,203 (6.4)
M518	Other specified intervertebral disc disorder	110,026 (5.2)
M545	Low back pain	106,841 (5.0)
M501	Cervical disc disorder with radiculopathy	77,730 (3.6)
M542	Cervicalgia	71,788 (3.4)
M480	Spinal stenosis	67,325 (3.2)
M509	Cervical disc disorder, unspecified	52,968 (2.5)

* The most common 10 disorders were extracted. ^†^ Counted as one disease code per admission regardless of the number of days hospitalised. ^‡^ Inpatients’ outpatient visits are also included. In total, 289,953 patients (inpatients, 47,965; outpatients, 241,988) were treated for a total of 2,682,258 times.

**Table 3 healthcare-10-00294-t003:** Adverse events associated with Chuna manipulation therapy.

Case	Total				Severity ^§^		
	*n*	Incidence 1 * (95% CI)	Incidence 2 ^†^ (95% CI)	Frequency	Mild	Moderate	Severe
				Classification ^‡^			
Increased musculoskeletal pain or discomfort	29	0.09 (0.06–0.12)	1.08 (0.73–1.52)	Very rare	9	19	1
Rib fracture	11	0.03 (0.02–0.06)	0.41 (0.21–0.70)	Very rare	0	11	0
Falls	6	0.02 (0.01–0.04)	0.22 (0.09–0.45)	Very rare	6	0	0
Dizziness	1	0.003 (0.000–0.013)	0.04 (0.00–0.16)	Very rare	0	1	0
Chest pain without rib fracture	2	0.01 (0.00–0.02)	0.07 (0.01–0.23)	Very rare	1	1	0
Mild discomfort	1	0.003 (0.000–0.013)	0.04 (0.00–0.16)	Very rare	1	0	0
Total	50	0.15 (0.11–0.19)	1.86 (1.39–2.43)	Very rare	17	32	1

* Incidence rate calculated with 100,000 person-days. ^†^ Incidence rate calculated with 100,000 treatment times. ^‡^ about incidence 1. Classified according to the European Commission guidelines for AEs of medicinal products (very common (≥1/10); common (≥1/100 to <1/10); uncommon (≥1/1000 to ≤1/100); rare (≥1/10,000 to ≤1/1000); very rare (≥1/10,000), and not known). ^§^ Classified according to National Cancer Institute. CI, confidence interval.

## Data Availability

We plan to share our findings with the participants, healthcare professionals, and the public through the publication of this report or trial registries. Data and materials can be requested by e-mail and will be provided after consultation with the IRB.

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
