# Peer review of "Safety of Chuna Manipulation Therapy in 289,953 Patients with Musculoskeletal Disorders: A Retrospective Study"

_healthcare, 2022, doi:10.3390/healthcare10020294_

Round 1

Reviewer 1 Report

Good morning.

Overall, I feel your paper is very well written and thoroughly well planned.

I have one minor comment:

  • Line 27- 2,682,259 treatments listed; line 151 2,682,258

Again, very well done and well written.

Reviewer 2 Report

Manuscript ID: 1550262

Manuscript title: Safety of Chuna Manipulation Therapy in 289,953 Patients with Musculoskeletal Disorders: A Retrospective Study

This study assesses multicenter data on the safety of Chuna manipulation therapy in patients at Korean medicine hospitals. The Introduction section presents a concise background and rationale for the study. The Methods section describes the procedures in detail and seems aligned to the study aims. The results are consistent with the study aims and methods, and discussion is limited to the reported findings. I have a few comments for the authors to consider.

Major comments

  1. I miss a ‘Statistical analysis’ section, although some information is indeed spread across the Results section. Consider grouping them in this section and reporting additional information, such as the calculation and reporting of uncertainties (e.g., 95% confidence intervals) for the point estimates of incidence and interrater agreement.

  1. Results. Considering that your sample includes a wide age range (0 to 80 or higher), it could be of interest to readers to show how the adverse events spread across ages. This should also be highlighted in sections 3.3.1 to 3.3.4 (notice that this is only mentioned in Discussion lines 249 and 262).

  1. Discussion. I wonder whether the authors consider as a possible limitation the lack of information about training and expertise level of the practitioners delivering the Chuna manipulation therapy.

Minor comments

  1. Abstract (lines 29-30). Please report uncertainties (e.g., 95% confidence intervals) for the point estimates of the incidence of adverse events.

  1. Methods (line 111). What was considered ‘inadequate data’? Please provide a few examples.

  1. Figure 1. If possible, consider reporting the number of patients and/or records screened at each stage of the research.

Reviewer 3 Report

2022 01 10_Kim_Chuna manip_healthcare_2022

I commend the authors on the completion of this manuscript.

The article includes a comprehensive introduction and background. This section is sufficient to demonstrate the justification for the development of the study in the field of knowledge.

The research question is well defined, being clinically relevant. The presentation defines the research question.

The research has been carried out in accordance with the ethical standards.

 But I have some relevant concerns highlighted below.

General comments

Minor grammar mistakes were observed in different parts of the paper. Could be good to check it prior to publishing.

References in the text must be before punctuation marks.

Specific Comments

  1. Materials and Methods

2.1. Study population and data collection. Please add some ethical considerations. How were the safety, right and privacy of the data treated?

Results

It would help a lot, if in “Table S3. Adverse events associated with increasing musculoskeletal pain”, Disease classification was changed to the name of the disease.

Line 170. Chest pain (n=2), correct to Chest pain (n=2; Table S6).

Line 172. Please, include the AE deemed to be serious.

Discussion

Line 326. Please, avoid speculation, especially if previous manuscripts are contradicted, for example regarding the recommendations on imaging tests, and the actual manuscript does not provide concrete data about it. In the actual manuscript, neither in the inclusion criteria, nor in the data collected from the electronic records, analysis of the imaging tests performed, before CMT, are provided. If it is possible, to discuss about this item it is better to include the data in the outcomes.

Conclusions

Please try to be more concrete according to the results in the conclusions section. Include incidence, for Mild, Moderate and Severe events. Describe the more frequent adverse events: Increased musculoskeletal pain or discomfort, Rib fracture, Falls. 

Round 2

Reviewer 3 Report

I commend the authors for the effort made to improve the manuscript and to carry out  the proposed changes.

Author Response

Thank you for your valuable advice. We have revised throughout the manuscript by native speaker.

This manuscript is a resubmission of an earlier submission. The following is a list of the peer review reports and author responses from that submission.

Round 1

Reviewer 1 Report

This is a retrospective study on adverse events in a very large cohort of patients undergoing Chuna manipulation. The strengths of this study are the large, multicentric cohort, and the evaluation of inter-rater agreement for the definition of adverse events. It is a well conducted study, although I think some methodological aspects could be clarified.

My main concern is the definition and criteria for retaining or excluding adverse events, which is a crucial methodological point. The authors ascertained the causality between treatment and adverse event as “probable, possible, or unlikely”, but then this classification is not reported. Were AEs excluded if unlikely? How many AEs were excluded, and why? Same question could be raised concerning comorbidities, which were analysed but no results were reported. Were comorbidities used as exclusion criteria? Or the other way around: did you find any association between comorbidities and AE?

I also struggle with the lack of control group, but I am not sure that it would be feasible in this context. I wonder if the authors have access to a cohort of patients who refused CMT, in order to compare the relative risks.

I have a few minor comments:

  • Material & Methods: you specify that comorbidities were analysed, but it is not clear what you did with this information. Was this an exclusion criterion? Did you find any association between comorbidities and AE?
  • Same comment about causality: how did you use the information about “probable, possible, or unlikely” causality?
  • Discussion: one difference between this study and previous ones, if I understand correctly, is that only highly trained physicians participated, and no chiropractor. Is this correct? You already conclude with “CMT performed by an expert can be considered relatively safe”, but I I think this should be further highlighted, since it is a good explanation of the low AE ratio compared to other studies. So the risk is not CMT: it is CMT performed by an unskilled person.

Reviewer 2 Report

The authors describe a large retrospective series of patients having benefited from Chuna Manipulation therapy and their potential outcome.

The lack of standardized protocols among healthcare providers, the heteregeneous FU and the several confounding factors and co-treatments prevent any conclusion to be made. Also one can only conclude about reported AE as it is easy to imagine that several unreported events may have occured and passed undiagnosed or lost to FU.

Unfortunately I do not think that this article brings any novelty to te field.